# Breast Cancer Beliefs and Screening Practices among Syrian Refugee Women and Jordanian Women

**DOI:** 10.3390/ijerph20043645

**Published:** 2023-02-18

**Authors:** Fatin Atrooz, Sally Mohammad Aljararwah, Chiara Acquati, Omar F. Khabour, Samina Salim

**Affiliations:** 1Department of Pharmacological and Pharmaceutical Sciences, College of Pharmacy, University of Houston, Houston, TX 77204, USA; 2Department of Medical Laboratory Sciences, Faculty of Applied Medical Sciences, Jordan University of Science and Technology, Irbid 22110, Jordan; 3Graduate College of Social Work, University of Houston, Houston, TX 77204, USA; 4Department of Clinical Sciences, Tilman J. Fertitta Family College of Medicine, University of Houston, Houston, TX 77204, USA; 5Department of Health Disparities Research, The University of Texas MD Anderson Cancer Center, Holcombe Blvd, Houston, TX 77030, USA

**Keywords:** breast cancer, Syrian refugees, Arab women, Jordan, mammogram, breast cancer screening

## Abstract

Despite significant declines in breast cancer (BC) incidence in the West, this disease is widespread in Jordan, where cancer detection occurs at much advanced stages. This is particularly concerning for Syrian refugee women resettled in Jordan, who are less likely to undergo cancer preventative procedures because of poor health literacy and lack of health services access. The present work assesses and compares breast cancer awareness and breast cancer screening behaviors among Syrian refugee women and Jordanian women residing close to the Syrian–Jordanian border city of *Ar-Ramtha*. A cross-sectional survey was conducted using a validated Arabic version of the Breast Cancer Screening Beliefs Questionnaire (BCSBQ). A total of 138 Syrian refugee women and 160 Jordanian women participated in the study. Results indicate that 93.6% of Syrian refugee women and Jordanian women ≥ 40 years of age reported never having undergone a mammogram. Syrian refugee women and Jordanian women reported low attitudes toward general health checkup (mean score for Syrian refugees 45.6 vs. 42.04 among Jordan women; *p* = 0.150). Barriers for BC screening were higher among Syrian refugees (mean score = 56.43) than Jordanian women (mean score = 61.99, *p* = 0.006). Women with higher education were more likely to report fewer barriers to screening (*p* = 0.027). The study documents a significant lack of BC screening awareness among Syrian refugee women and Jordanian women, indicating that future work is needed to alter current attitudes towards mammograms and early detection measures especially for Syrian refugee women and Jordanian women residing in rural areas of Jordan.

## 1. Introduction

Despite increased awareness, availability of better diagnostic tools, and improved therapeutic interventions, breast cancer (BC) remains the most diagnosed cancer and the second-leading cause of cancer death among women [1]. This is particularly concerning in the Middle East region where the incidence of breast cancer among Arab women has significantly increased compared to women in developed Western countries [2]. Recent data indicate that Arab women develop breast cancer at an early median age of 43–52 years, with an average age of 48 at the time of diagnosis (SD = 2.8) [3], while the literature has reported a median age of 62 among women from the US [4]. Furthermore, breast cancer diagnosis occurring at an early age among Arab women may be the result of an interplay between environmental and genetic factors [5]. A significantly higher risk of mortality exists among Arab women primarily due to the advanced stage of the cancer at the time of diagnosis [6]. According to the data collected in 2018 by the Jordan Cancer Registry, breast cancer was the most common (38.9%) of all cancer types, and it was reported to be the main cause of cancer-related death among females in Jordan (15.7%) [7]. Similarly, breast cancer was most commonly detected among female Syrian refugees residing in Jordan (40 [36%] of 112) during 2015–2017 according to data reported by the UNHCR [8]. Thus, despite a significant decline in the incidence of breast cancer in the West, breast cancer in Jordan is on the rise [9]. While BC screening will not impact the incidence of breast cancer, early detection of the disease may enhance/contribute to more timely treatment and enhanced survival rates [10]. The Jordan Breast Cancer Program (JBCP), established in 2007, has played a significant role in coordinating breast cancer early detection efforts across Jordan and in providing screening services to Jordanian women [9]. Despite multiple initiatives to raise breast cancer awareness and promote breast cancer screening, women in Jordan continue to have limited knowledge about breast cancer and exhibit low participation rates in breast cancer screening and early detection programs [7,11]. A significant portion of Syrian refugees who reside within host communities in Jordan have access to public health services, including breast cancer screening. Healthcare costs for Syrian refugees are comparable to those of uninsured Jordanians [12]. On the other hand, refugees living in camps have free access to healthcare services provided by the Jordanian government and international refugee assistance agencies, which tend to be in most cases suboptimal [13]. The private health sector in Jordan offers a wide range of health services, including breast cancer screening; however, financial constraints are significant/important barriers to accessing these services [14].

Information about the extent of breast cancer knowledge, attitudes, and barriers toward breast cancer screening among Jordanian women is limited. Despite the limited evidence available to date, findings have consistently revealed inadequate breast cancer knowledge and elevated/high barriers to cancer screening among women. The literature has often cited extreme religiosity, high fatalism, patriarchal culture, stigma, and other socioeconomic factors among the number of characteristics/factors that contribute to this gap in knowledge/awareness, in addition to low participation rates in early detection practices [15]. This is further complicated by a complex population landscape of Jordan with presence of a sizable Syrian refugee women population resettled in Jordan. An estimated 1.4 million Syrians live in Jordan, of which over 680,000 are registered refugees [16]. Given the limited number of studies conducting on BC screening attitudes in the region, the present contribution was designed to explore breast cancer awareness, beliefs, and attitudes towards breast cancer screening among Syrian refugee women and native Jordanian women in the northern city of *Ar-Ramtha*, which is situated on the Syrian–Jordanian border.

## 2. Materials and Methods

All communication forms and survey questionnaires utilized in the study were approved by the Institutional Review Board (IRB) Committee (STUDY00002929) for the Protection of Human Subjects at the University of Houston (UH), Houston, TX, USA, and the Jordan University of Science and Technology (JUST) IRB, Irbid, Jordan (IRB#52/148/2022, 10.05.2022).

### 2.1. Subject Recruitment

Upon approval of the study protocol by the UH-IRB and JUST-IRB committees, Syrian refugee women and Jordanian women residing in the city of *Ar-Ramtha* (Jordan) were recruited to participate in the study. *Ar-Ramtha* is a small city in the northern part of Jordan compromising 7 small villages. The population of *Ar-Ramtha* is about 240,000. The number of Syrian refugees who have settled in *Ar-Ramtha* is estimated to be approximately 40,000 [17]. Both Syrian refugee women and Jordanian women were recruited from the same residential areas. Convenient sampling and snowball recruitment methods were utilized through local Jordan community networks. Snowball sampling was achieved by requesting consenting study participants to refer new potential participants from their social network to the investigative team. Recruitment continued until the target number (about 300 participants) was achieved. Women were included in the study if they: (1) were adult (18 years or older) Syrian refugee women or Jordanian women; (2) identified as women; (3) spoke the Arabic language. The postdoctoral fellow (F.A.) and the student researcher (S.M.A.) involved in the study are native Arabic speakers who explained the study objectives to the participants in Arabic (native language of participants and obtained participant consent. Arabic versions of the survey questionnaires were uploaded on the REDCap platform, which enables secure building and management of online surveys. The study team used the survey link to access the questionnaires while interviewing the participants. Participants had the choice to complete the survey either independently or with some one-on-one guidance by the study team. Upon survey completion, each participant received a USD 14 (10 Jordanian Dinar, JD) gift card.

### 2.2. Measures

The survey included a sociodemographic questionnaire assessing age, education level, marital status, number of children, number of families living in the same household, health insurance status, employment status, and average monthly income. Questions related to body mass index (weight and height), tobacco use, and prevalence of chronic diseases (diabetes, hypertension, hypothyroidism, asthma, irritable bowel syndrome, and cancer) were also included in the demographic characteristics questionnaire.

A validated Arabic version of Breast Cancer Screening Beliefs Questionnaire (BCSBQ) [10] was used to evaluate breast cancer awareness and screening barriers among Syrian refugee women and Jordanian women. BCSBQ assesses knowledge and attitudes towards breast cancer and screening practices of respondents [18,19]. The BCSBQ is a 13-item instrument composed of three subscales: (1) attitude toward health checkups, which includes four items designed to ascertain whether a woman had general health checkups in the absence of signs and symptoms; (2) knowledge and perceptions about breast cancer (four items), which explored cultural beliefs related to breast cancer; and (3) barriers to screening practices of mammograms (five items), including personal and practical issues perceived by women which prevent them from undergoing breast cancer screening. All 13 items were rated on a 5-point Likert scale ranging from ‘strongly agree’ (1) to “strongly disagree” (5). Lower scores indicate the lowest attitudes, least knowledge, or greatest barriers [18,19,20]. The research team included all the items in the original BCSBQ version for assessing the attitude toward health checkup subscale and knowledge about breast cancer subscale. For the barrier subscale, the item that asks participants if they cannot have mammogram screening because they cannot speak English, in the original version, is not applicable in Jordan; therefore, this item was replaced with the following item “we can’t perform mammogram because we can’t afford the cost”. The BCSBQ instrument showed adequate reliability in our sample, with a Cronbach’s alpha coefficients of 0.92 in the attitudes towards general health checkups subscale, 0.70 in the knowledge and perceptions about breast cancer subscale, and 0.72 in the barriers to screening practices of mammograms subscale.

### 2.3. Body Mass Index (BMI) Calculation

Body weight, reported in kilograms (Kg), and the height, reported in meters (m), were utilized to calculate the body mass index (BMI) using the following formula:BMI = Weight (Kg)/[height (m)]^2^

BMI values were interpreted using standard weight status categories suggested for adult women and men: <18.5 = underweight, 18.5–24.9 = normal weight, 25–29.9 = overweight, and 30+ = obese [21].

### 2.4. Data Analysis

Descriptive statistics were used to summarize demographic characteristics of the participants and their scores in the BCSBQ subscales. Factors associated with performing screening practices (clinical breast exam and mammogram) by age groups were investigated by comparing the proportion of refugees and Jordanian women within each category. Sample comparisons between Syrian refugee women and Jordanian women were performed using independent *t*-test for continuous variables, or chi-square test for categorical variables. Correlation analysis between breast cancer screening practices and participants’ sociodemographic characteristics was conducted using Spearman test. Linear regression analysis was used to analyze predictors of BCSBQ subscales scores. Significance was set at *p* < 0.05. All analyses were conducted using IBM SPSS software, version 29.0.

## 3. Results

The present cross-sectional study was conducted between 14 July and 1 October 2022 among Syrian refugee women and Jordanian women who resided in the *Ar-Ramtha* area in northern Jordan. A total of 298 women including 138 Syrian refugee women (46%) and 160 Jordanian (54%) were included in the analysis. The response rate of the participants was in the range of 97.3–99.7% for the two groups.

### 3.1. Demographic and Socioeconomic Characteristics of the Participants

As shown in Table 1, age distribution was comparable between Syrian refugee women and Jordanian women (*p* = 0.109). The age of the participants ranged from 18 years to 77 years, with an average age of 41.38, SD = 13.39 years. Approximately 75.5% of Syrian refugee women and Jordanian women were married, with no difference between the two groups (*p* = 0.304). However, Syrian refugee women reported having more children in the categories of (3–7) and (8–13) and fewer children in the categories (0) and (1–2) as compared to Jordanian women (*p* < 0.001), suggesting that Syrian refugee women had larger families at time of study completion. There was a significant difference in the education level between Syrian refugee women and Jordanian women (F_3,278_ = 43.451, *p* < 0.001), with 84.9% (N = 118) of Syrian refugee women reporting not having completed high school as compared to 48.8% (N = 78) of Jordanian women.

Approximately 97.8% of Syrian refugees (N = 135) reported a monthly income below JD 551 (~USD 750), which is significantly higher than the 81.9% (N = 131) of Jordanian women who reported a monthly income < JD 551 (~USD 750) (*p* < 0.001). The percent of Syrian refugee women who reported living with more than one family member in the same household (N = 54, 38.8%) was significantly (*p* < 0.001) higher than Jordanian women (N = 17, 10.6%). Most of the Jordanian women (81.9%; N = 131) reported having health insurance, versus only 6.5% (N = 9) of Syrian refugee women (*p* < 0.001).

According to the calculated body mass index (BMI), only 27.6% (N= 80) were considered within the normal healthy weight category, and 39.0% were obese (N = 113). No difference in the percentage of Syrian refugee women and Jordanian women within the BMI categories (*p* = 0.757) were noted. The prevalence of chronic diseases was comparable among Syrian refugee women and Jordanian women, with hypertension (20.8%, N = 62), hypothyroidism (20.8%, n = 62), and irritable bowel syndrome (22.1%, n = 66) being the most prevalent diseases; see Table 1.

Tobacco use was reported by both groups, with 13.8% (n = 19) of Syrian refugee women and 13.1% (n = 21) of Jordanian women reporting smoking cigarettes (*p* = 0.871). Interestingly, a higher percentage of Jordanian women (25%, n = 40) reported waterpipe use as compared to Syrian refugees (14.5%, n = 20; *p* = 0.024). See Table 1.

### 3.2. Breast Cancer History and Screening Practices among Participants

Two Syrian refugee women (1.4%) and one Jordanian woman (0.6%) reported having a history of breast cancer; see Table 2. Around 5.8% (n = 8) of Syrian refugee women and 2.5% (n = 4) of Jordanian women reported that someone from their immediate family had breast cancer with no significant difference between the two groups (*p* = 0.205). Jordanian women reported significantly higher rates of breast self-exam compared to Syrian refugee women: 15.6% of Jordanian women reported performing breast self-exams at least once a month compared to 8.7% of the Syrian refugee women. Additionally, 31.9% of Jordanian women reported performing self-exams once or twice a year compared to 15.9% of Syrian refugee women (*p* < 0.001), see Table 2.

In Jordan, women are recommended to start clinical breast exam at the age of 25. Accordingly, we examined the rate of clinical exam among women 25 years and older. The rate of clinical breast exam was low among both groups of women, with 82.4% of the women ≥ 25 years old reporting never having done a clinical breast exam, and only 4.2% of women performed clinical breast exam within the past year. No significant difference was observed between Syrian refugee women and Jordanian women in the rate of clinical breast exam (*p* = 0.178); see Table 2.

Annual mammogram screening is recommended for women aged 40 or older in Jordan. In our sample, 93.6% of Syrian refugee women and Jordanian women who were 40 years or older reported never having undergone mammogram screening. Only 2.6% of Syrian refugee women and 4.3% of Jordanian women 40 years and older reported having annual mammogram screenings. No statically significant differences were detected between the two groups (*p* = 0.806); see Table 2.

### 3.3. Correlation Analysis between Breast Cancer Screening Practices and Sociodemographic Characteristics

Correlation analysis, as shown in Table 3**,** indicated that Syrian refugee women are less likely to perform breast self-exam as compared to Jordanian women (*p* < 0.001). Married, widowed, or divorced women were more likely to perform breast-self exams than single women (*p* < 0.001). Women with monthly income less than JD 551 and women living with more than one family in the same household were less likely to perform breast self-exam compared to women with monthly income > JD 551 (*p* = 0.001) who did not live with more than one family in the same household (*p* = 0.021). Women with education levels higher than high school and with health insurance were more likely to perform breast self-exams as compared to women with education level less than high school (*p* < 0.001) and without health insurance (*p* < 0.001), see Table 3.

For clinical breast exams, women who were older than 25 years of age and women with health insurance were more likely to perform clinical breast exams than those without health insurance (*p* = 0.032), while women with monthly income below 551 were less likely to perform clinical breast exams than women with monthly incomes more than JD 551 (*p* = 0.009); see Table 3.

For adherence to mammogram screening among women 40 or older, women who had a history of breast cancer were more likely to perform mammograms when compared to women who did not have history of breast cancer (*p* < 0.001), while women with children were less likely to perform mammograms as compared to women with no children (*p* = 0.016), see Table 3.

Syrian refugee women and Jordanian women reported low attitudes toward general health checkups with average scores of 45.65 in Syrian refugee women and 42.04 in Jordanian women (*p* = 0.150). Regression analysis revealed that Syrian refugee women who were 25 and older were more likely to exhibit higher attitudes toward general health checkup than Jordanian women (*p* = 0.002) or women younger than 25 years (*p* = 0.036), while those with monthly income less than JD 551 were less likely to report an elevated attitude toward health checkups (*p* < 0.001), see Table 4.

Knowledge about breast cancer was comparable between Syrian refugee women and Jordanian women, with reported average scores of 56.67 and 59.78, respectively (*p* = 0.131). Women with education levels higher than high school and who had a family history of breast cancer were more likely to exhibit higher knowledge about breast cancer as compared to women with lower education background/level (*p* < 0.001) or women with no family history of breast cancer (*p* = 0.031); see Table 4.

Barriers to mammogram screening were higher among Syrian refugee women (mean score = 56.43) than Jordanian women (mean score = 61.99, *p* = 0.006). Women with high school or higher education levels were more likely to exhibit fewer barriers for mammograms as compared to women with education levels less than high school (*p* = 0.027). Women with low monthly income (less than JD 551) were more likely to experience a greater number of barriers for undergoing mammograms than those with monthly income over JD 551 (*p* = 0.046); see Table 4.

## 4. Discussion

In this cross-sectional study, we assessed breast cancer knowledge, attitudes, and barriers towards mammography screening among Syrian refugee women and Jordanian women residing in a rural area in northern Jordan. Our data show that the rate of clinical breast exam and annual mammogram screening were very low among Syrian refugee women and Jordanian women, with 82.4% of the women ≥ 25 years old reporting never having undergone a clinical breast exam, and 93.6% of women who were 40 years or older reporting never having undergone mammogram screening. Therefore, there is a critical need for improving awareness and attitudes toward breast cancer screening among Syrian and Jordanian women residing in the rural areas in Jordan. Despite the Jordanian government’s initiatives to raise breast cancer awareness and promote breast cancer screening, women in Jordan continue to have limited knowledge about breast cancer and exhibit low participation rates in breast cancer screening and early detection, particularly women who live in rural areas of Jordan. Therefore, it is warranted that BC awareness programs must amplify outreach in the rural areas. For example, home visits and social media messaging may be an effective outreach strategy for improving knowledge about breast health and for promoting BC screening practices among women in the rural areas.

Although Syrian refugee women and Jordanian women demonstrated comparable knowledge about breast cancer, Jordanian women reported significantly higher rates of breast self-exam than Syrian refugee women. Although the effectiveness of breast-self exam on early BC detection is debatable [22,23,24], breast self-exam is a well-established indicator of BC awareness and positive attitudes toward screening beliefs. Health insurance was one of the predictors of breast self-exams (see Table 3), which suggests that Jordanian women may access family clinics more frequently than Syrian refugee women, where they learn healthy practices such as breast-self exams. Women in the low-income category (< JD 551 /month) were less likely to practice breast self-exams or clinical exams, suggesting that the health care cost constitutes a major barrier for breast health screening among Syrian refugee women. Several studies have suggested that Syrian refugee women have limited access to reproductive health care and family planning clinics in Jordan [25,26]. Our data also suggest that more educated women were more likely to perform breast self-exams than women who had lower levels of education, further highlighting the issue of limited enrollment of Syrian refugee women in the Jordanian education system [27], which not only impacts their general knowledge and employment opportunities but also affects their health practices and self-care.

Barriers for mammogram screening were significantly higher among Syrian refugee women as compared to Jordanian women; see Table 4. Women with breast cancer history were more likely to perform mammogram screenings as these should be part of their follow-up treatment plan. However, women with a family history of breast cancer did not report higher screening rates (see Table 3) despite reporting higher knowledge about breast cancer; see Table 4. The data is concerning as women with family history of breast cancer have higher risk of developing the disease or for carrying this predisposition for future generations [28,29].

Women with limited education and income (<JD 551) were more likely to experience greater barriers to mammogram screening. This finding is similar to those of previous works conducted in Jordan [7,11] and other Arab countries [30,31]. Low levels of education could negatively affect the screening practices. This may be due to limited health literacy or affected/impaired communication with health-care professionals. Overall, results indicate that future breast cancer educational programs should focus on women with lower levels of education and/or or lower socioeconomic status, including the community of Syrian refugee women. It is our recommendation that policy makers consider addressing screening barriers identified in this study, such as the access to the health systems, the cost, and most importantly, education, particularly among the refugee women.

## 5. Limitations

Our study has several limitations. The first is the reliance on cross-sectional and self-report data as opposed to clinical interview and/or retrospective data analysis from cancer centers and hospital. The second is the small sample size focused on one city. Therefore, the results cannot be generalized beyond the context of the present survey. Additionally, our sample also included women less than 25 years old (16%), which is in general not a targeted breast cancer screening group according to prevailing screening standards [32].

## 6. Conclusions

Breast cancer knowledge, attitude, beliefs, and screening rates are very limited among Syrian refugee women and Jordanian women who reside in the rural areas of northern Jordan. Therefore, there is a critical need for improving awareness and attitudes toward breast cancer screening, particularly for women with low education levels and low socioeconomic status. Future research can target current limitations in screening awareness by implementing educational programs that highlight the importance of breast cancer screening for early detection. The present work also has implications for current cancer screening policies and the need to address factors that may act as barriers for timely access to preventative care.

## Figures and Tables

**Table 1 ijerph-20-03645-t001:** Demographic characteristics and chronic diseases prevalence in Syrian refugee women and Jordanian women.

Variable	Syrian Refugee Women (n = 138) N (%)	Jordanian Women (n = 160) N (%)	Total (n = 298) N (%)	Statistics	*p* Value
**Age**					
18–25	27 (20.6)	19 (12.4)	46 (16.2)	6.053	0.109
26–39	33 (25.2)	44 (28.8)	77 (27.1)		
40–59	64 (48.9)	73 (47.7)	137 (48.2)		
60+	7 (5.3)	17 (11.1)	24 (8.5)		
**Relationship Status**					
Single	9 (6.5)	17 (10.6)	26 (8.7)	3.635	0.304
Married	103 (74.6)	122 ((76.3)	225 (75.5)		
Divorced	9 (6.5)	5 (3.1)	15 (4.7)		
Widowed	16 (10.0)	17 (12.3)	33 (11.1)		
**Number of Children**					
0	9 (6.6)	47 (30.1)	56 (19.2)	38.089	<0.001 **
2–1	26 (19.1)	45 (28.8)	71 (24.3)		
7–3	95 (69.9)	61 (39.1)	156 (53.4)		
13–8	6 (4.4)	3 (1.9)	9 (3.1)		
**Education Level**					
Less than High School	117 (84.8)	78 (48.8)	195 (65.4)	42.959	<0.001 **
High School	13 (9.4)	44 (27.5)	57 (19.1)		
College	8 (5.8)	37 (23.1)	45 (15.1)		
Graduate Studies	0 (0.0)	1 (0.3)	1 (0.3)		
**Employment**					
Yes	26 (18.8)	32 (20.3)	58 (19.6)	0.093	0.76
No	112 (80.4)	126 (79.7)	238 (80.4)		
**More than one family lives in the same household**					
Yes	54 (39.1)	17 (10.6)	71 (23.8)	33.173	<0.001 **
No	84 (60.9)	143 (89.4)	227 (76.2)		
**Average Monthly Income**					
Less than JD 250	92 (67.2)	55 (34.4)	147 (49.5)	39.684	<0.001 **
JD 250–550	42 (30.7)	76 (47.5)	118 (39.7)		
JD 551–800	3 (2.2)	14 (8.8)	17 (5.7)		
JD 801–1500	0 (0.0)	11 (6.9)	11 (3.7)		
More than JD 1500	0.(0.0)	4 (2.5)	4 (1.3)		
**Health Insurance**					
Yes	9 (6.5)	131 (81.9)	190 (53.0)	168.902	<0.001 **
No	129 (93.5)	29 (18.1)	158 (53.0)		
**Body Mass Index**					
Underweight (<18.5)	2 (1.5)	5 (3.2)	7 (2.4)	1.185	0.757
Normal Weight (18.5–24.9)	36 (26.9)	44 (28.2)	80 (27.6)		
Overweight (25–29.9)	44 (32.8)	46 (29.5)	90 (31.0)		
Obese (30+)	52 (38.8)	52 (38.8)	113 (39.0)		
**Diabetes**					
Yes	20 (14.5)	23 (14.4)	43 (14.4)	0.001	0.977
No	118 (85.5)	137 (85.6)	225 (85.6)		
**Hypertension**					
Yes	30 (21.7)	32 (20.0)	62 (20.8)	0.136	0.712
No	108 (78.3)	128 (80.0)	236 (79.2)		
**Hypothyroidism**					
Yes	8 (5.8)	11 (6.9)	19 (6.4)	0.144	0.704
No	130 (94.2)	149 (93.1)	279 (93.6)		
**Asthma**					
Yes	8 (5.8)	5 (3.1)	13 (4.4)	1.268	0.26
No	130 (94.2)	155 (96.9)	285 (95.6)		
**Irritable Bowel Syndrome**					
Yes	31 (22.5)	35 (21.9)	66 (22.1)	0.015	0.903
No	107 (77.5)	125 (78.1)	232 (77.9)		
**Smoking Cigarettes**					
Yes	19 (13.8)	21 (13.1)	40 (13.4)	0.026	0.871
No	119 (86.2)	139 (86.9)	258 (86.6)		
**Waterpipe Use**					
Yes	20 (14.5)	40 (25.0)	60 (20.1)	5.087	0.024 *
No	118 (85.5)	120 (75.0)	238 (9.9)		

JD: Jordanian Dinar (USD 0.70). * *p* < 0.05; ** *p* < 0.01.

**Table 2 ijerph-20-03645-t002:** Breast cancer history and screening practices among Syrian refugee women and Jordanian women.

Variable	Syrian Refugee Women N (%)	Jordanian Women N (%)	Total N (%)	Statistics	*p* Value
*Have a History of Breast Cancer—All Participants*					
Yes	2 (1.4)	1 (0.6)	3 (1.0)	0.497	0.481
No	136 (98.6)	158 (99.4)	294 (99.0)		
(Total)	138 (46.5)	159 (53.5)	297 (100.0)		
Someone From Immediate Family Had Breast Cancer—All Participants					
Yes	8 (5.8)	4 (2.5)	12 (4.1)	3.165	0.205
No	129 (93.5)	153 (97.5)	282 (95.6)		
I don’t Know	1 (0.7)	0 (0.0)	1 (0.3)		
(Total)	138 (46.8)	157 (53.2)	295 (100.0)		
*How Often Is Breast Self-Exam Performed? All Participants*					
Never	104 (75.4)	84 (52.5)	188 (63.1)	16.683	<0.001 **
Once or Twice/Year	22 (15.9)	51 (31.9)	73 (12.4)		
At Least Once/Month	12 (8.7)	25 (15.6)	37 (12.4)		
(Total)	138 (46.3)	160 (53.7)	298 (100.0)		
*When Was Your Last Clinical Breast Exam? Target Age Group ≥ 25*					
Never	102 (87.2)	113 (78.5)	215 (82.4)	3.453	0.178
More than a Year	11 (9.4)	24 (16.7)	27 (15.7)		
A Year or Less	4 (3.4)	7 (4.9)	11 (4.2)		
(Total)	117 (44.8)	144 (55.2)	261 (100.0)		
*How Often Do You Have a Mammogram? Target Age Group ≥ 40*					
Never	74 (94.9)	87 (92.6)	161 (93.6)	0.432	0.806
Once Every 2–3 Years	2 (2.6)	3 (3.2)	5 (2.9)		
Once a Year	2 (2.6)	4 (4.3)	6 (3.5)		
(Total)	78 (45.3)	94 (54.7)	172 (100.0)		
*How Many Mammograms Have You Had in the Last 5 Years? Target Age Group ≥ 40*					
0	74 (94.9)	89 (94.7)	163 (94.8)	0.035	0.983
1	3 (3.8)	4 (4.3)	7 (4.1)		
2	1 (1.1)	1 (1.3)	2 (1.2)		
(Total)	78 (45.3)	94(54.7)	87 (100.0)		

Note: ** *p* < 0.01.

**Table 3 ijerph-20-03645-t003:** Correlation analysis between breast cancer screening practices and participants’ sociodemographic characteristics.

Variable	Statistics	Breast Self-Exam (Ref Never) All Participants N = 298	Clinical Breast Exam (Ref Never) Target Group > 25 N = 261	Mammogram (Ref Never) Target group > 40 N = 172
Syrian Refugee Women (ref Jordanian Women)	*r*	−0.236 **	−0.114	−0.047
	*p*	<0.001	0.067	0.539
History of Breast Cancer (ref no)	*r*	−0.077	0.044	0.328 **
	*p*	0.183	0.477	<0.001
Family History of Breast Cancer (ref no)	*r*	0.020	−0.040	−0.045
	*p*	0.729	0.521	0.556
Relationship Status (ref Single)	*r*	0.138 *	−0.025	−0.0126
	*p*	0.017	0.685	0.100
Children (ref no Children)	*r*	−0.077	0.000	−0.183 *
	*p*	0.185	0.996	0.016
Average Monthly Income (ref > JD 551 (~USD 750)	*r*	−0.186 **	−0.161 **	−0.066
	*p*	0.001	0.009	0.390
Education (ref Less than High School)	*r*	0.206 **	0.092	−0.102
	*p*	<0.001	0.138	0.185
Employment (ref no)	*r*	0.050	0.031	0.073
	*p*	0.390	0.615	0.343
More than One Family Lives in the Same Household (ref no)	*r*	−0.134 *	−0.069	−0.047
	*p*	0.021	0.266	0.536
Health Insurance (ref no)	*r*	0.227 **	0.133 *	0.030
	*p*	<0.001	0.032	0.697

Data were analyzed using Spearman test. JD: Jordanian Dinar (USD 0.70); * *p* < 0.05; ** *p* < 0.01; 3.4. Breast Cancer Screening Beliefs Questionnaire (BCSBQ) Subscales

**Table 4 ijerph-20-03645-t004:** BCSBQ subscales and linear regression for significant predictors of subscales scores.

Variable	Syrian Refugee Women (n = 138) Mean (SD)	Jordanian Women (n = 159) Mean (SD)	*p*-Value	Significant Predictors	β	SE	*p*-Value
Attitude toward health checkups	45.65 (20.70)	42.04 (22.10)	0.150	Income (ref > 551 JD)	−0.272	4.109	<0.001 ***
				Syrian Refugee Women (ref Jordanian Women)	0.183	2.499	0.002 **
				Age (ref less than 25 years)	0.121	3.761	0.036 *
Knowledge about breast cancer	56.67 (16.25)	59.78 (18.81)	0.131	Education (ref less than high school)	0.302	2.177	<0.001 ***
				Family History of Breast Cancer (ref no)	0.122	4.928	0.031 *
Barriers to mammogram screening	56.43 (15.26)	61.99 (18.76)	0.006 **	Income (ref > 551 JD)	−0.126	3.637	0.046 *
				Education (ref less than high school)	0.140	2.419	0.027 *

Data were analyzed using t-tests for scores comparison and linear regression for score predictors. Note: Lower scores in attitude, knowledge, and barriers to mammogram screening subscales represent lower attitudes, less knowledge, and greater barriers, respectively. JD: Jordanian Dinar (USD 0.70). * *p* < 0.05; ** *p* < 0.01; *** *p* < 0.001.

## Data Availability

The data presented in this study are available on request from the corresponding author. The data are not publicly available due to privacy.

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
