# Peer review of "Breast Cancer Beliefs and Screening Practices among Syrian Refugee Women and Jordanian Women"

_ijerph, 2023, doi:10.3390/ijerph20043645_

Round 1
Reviewer 1 Report
This is a well-written and well-designed study that addresses an issue of importance in the context of breast cancer and refugee health. We have the following concerns that we ask to be addressed:
1. The abbreviation BC (Breast Cancer) seems to be used irregularly, in the abstract and also in the body of the manuscript. For example, it first appears on line 18, but then on line 25 the authors use "Breast Cancer".
2. I would put more emphasis and elaborate on how the barriers can be addressed. For example, elaborate more on how the government and refugee agencies can help.
3. On line 156, the opening of "("/parenthesis is missing.
Author Response
RESPONSE LETTER
We are thankful to the reviewers for their thoughtful and detailed review of our manuscript. All the recommended changes and edits have been incorporated in the revised version of our manuscript. We greatly appreciate the time and effort invested to produce this detailed review. Our point-by-point response is provided below.
This is a well-written and well-designed study that addresses an issue of importance in the context of breast cancer and refugee health. We have the following concerns that we ask to be addressed:
- The abbreviation BC (Breast Cancer) seems to be used irregularly, in the abstract and also in the body of the manuscript. For example, it first appears on line 18, but then on line 25 the authors use "Breast Cancer".
Response: We apologize for lack of clarity. The abbreviation BC in the abstract section is referred to Breast Cancer, while the BCSBQ is used for Breast Cancer Screening Beliefs Questionnaire which was used for evaluation of breast cancer knowledge and screening practices among participants.
- I would put more emphasis and elaborate on how the barriers can be addressed. For example, elaborate more on how the government and refugee agencies can help.
Response: This is an excellent point raised by the reviewer. Based upon this recommendation, we have now included some discussion on the value of amplifying breast cancer awareness programs and drawing attention for policy makers and healthcare providers (please note first and last paragraph in the revised manuscript).
- On line 156, the opening of "("/parenthesis is missing.
Response: Thank you, the opening parenthesis has been added in the revised manuscript.
Reviewer 2 Report
A very well written paper of a survey-based cross-sectional study examining breast cancer beliefs and screening practices among Syrian refugee and Jordanian women.
Detailed comment:
· Lack of sufficient information in Results: Have all the 138 Syrian refugee women and 160 Jordanian women answered every single questions and if not, what have you done to deal with the missing data?
· Text in 3.1 is a bit redundant since table 1 give a very clear picture of demographic data. Suggest to remove some of the sentences to make this section more concentrated.
· Line 246: the authors stated the ANCOVA analysis controlled for age, how come the age became a predictor?
· 2.4 and 3.2: It would be good to use consistent terminology for statistical test, ANCOVA or liner regression.
· Lines 267-273, it is more suitable to be placed in the background.
· There are minor typos, grammar errors (e.g. line 279, suggests) and format errors (e.g in-text citation [4] in line 49).
· Authors could enlarge the part of the implications of their study findings in breast screening programs/practice in Jordan. Because this topic was only touched very basically in lines 313-315 and 320-324.
· Limitations should be placed before Conclusions.
Author Response
RESPONSE LETTER
We are thankful to the reviewers for their thoughtful and detailed review of our manuscript. All the recommended changes and edits have been incorporated in the revised version of our manuscript. We greatly appreciate the time and effort invested to produce this detailed review. Our point-by-point response is provided below.
- A very well written paper of a survey-based cross-sectional study examining breast cancer beliefs and screening practices among Syrian refugee and Jordanian women.
Response: Thank you.
- Detailed comment:
Lack of sufficient information in Results: Have all the 138 Syrian refugee women and 160 Jordanian women answered every single question and if not, what have you done to deal with the missing data?
Response: We regret lack of clarity on this issue. As recommended by the reviewer, we have now added response rate and details of missing data in the revised manuscript. Response rate was 99.7%, the missing data was insignificant. The women answered all the questions. This information has been included.
- Text in 3.1 is a bit redundant since table 1 give a very clear picture of demographic data. Suggest to remove some of the sentences to make this section more concentrated.
Response: Done as recommended.
- Line 246: the authors stated the ANCOVA analysis controlled for age, how come the age became a predictor?
Response: We apologize for this error. The sentence about ANCOVA analysis was a carry-over from a previous draft and has removed. In this study we have used independent t-test for continuous variables and chi-square test for categorical variables.
- 2.4 and 3.2: It would be good to use consistent terminology for statistical test, ANCOVA or liner regression.
Response: Linear regression was used to analyze predictors of BCSBQ subscales scores. ANCOVA test was removed from the revised manuscript.
- Lines 267-273, it is more suitable to be placed in the background.
Response: This is an excellent suggestion. We agree with the reviewer and as suggested we have moved the mentioned paragraph to the introduction.
- There are minor typos, grammar errors (e.g. line 279, suggests) and format errors (e.g in-text citation [4] in line 49).
Response: The highlighted grammatical typing errors were fixed.
- Authors could enlarge the part of the implications of their study findings in breast screening programs/practice in Jordan. Because this topic was only touched very basically in lines 313-315 and 320-324.
Response: Done as recommended in the first and last paragraphs of the discussion.
- Limitations should be placed before Conclusions.
Response: Done as recommended.
Reviewer 3 Report
This is a study dealing with BC awareness and screening adherence btw rural Jordanian and Syrian refugee’s population according to self-reported questionnaires. The topic is very important and relevant, the paper is well written. I do have a few notes regarding the design and power of this study:
In the Introduction:
- Lack of awareness and screening itself will not lead to increased incidence. What is the explanation to it?
- The younger age of diagnosis compared to US can also be attributed to the general life expectancy differences btw the countries.
- I suggest the authors add a few words in the introduction about the current medical system (public vs private etc) in Jordan and particularly the BC screening recommendations/ coverage offered to the general population, if any, and whether this program applies same btw Jordan citizens and Syrian refugees.
- Methods- can the authors better described how the cohort has been collected- was it pure random sample of individuals? What do they mean by snowball?
- Results- Table 1- given the large numbers of births it’s a bit surprising that so few live with more than one family member.
- Discussion- obesity does slightly increase the risk for BC, as well as multi-parity, however the Syrian refugees who had more children performed less self-exam. Therefore I would suggest to not link btw BC risk factors and adherence to screening. (Same applies for smoking at the end of the paragraph).This study does not have the power to link the 2.
- Breast self-exam has been proved of not being an effective method of BC early detection, it can merely serve here as a surrogate for general awareness. This might be worth mentioning in the paper.
- To the limitations of the study I suggest adding the fact that the average age was quite low, as women under 30, and especially under 25 are largely not the target population for BC screening, according to most European and American guidelines.
- Unfortunately the full questionnaire was not part of the review. It’s interesting to see what were the barriers for screening questions.
Author Response
RESPONSE LETTER
We are thankful to the reviewers for their thoughtful and detailed review of our manuscript. All the recommended changes and edits have been incorporated in the revised version of our manuscript. We greatly appreciate the time and effort invested to produce this detailed review. Our point-by-point response is provided below.
This is a study dealing with BC awareness and screening adherence btw rural Jordanian and Syrian refugee’s population according to self-reported questionnaires. The topic is very important and relevant, the paper is well written. I do have a few notes regarding the design and power of this study:
In the Introduction:
- Lack of awareness and screening itself will not lead to increased incidence. What is the explanation to it?
Response: We agree with the reviewer. Thanks for raising this issue. We have now rephrased this part by focusing the discussion on the impact of breast cancer and early detection rather than increased incidence.
- The younger age of diagnosis compared to US can also be attributed to the general life expectancy differences btw the countries.
Response: The breast cancer diagnosis occurring at a younger age among the Syrian and Jordanian population may be a result of an interplay between environmental and genetic factors and may not have a bearing on life expectancy issues [PMID: 33671879].
- I suggest the authors add a few words in the introduction about the current medical system (public vs private etc) in Jordan and particularly the BC screening recommendations/ coverage offered to the general population, if any, and whether this program applies same btw Jordan citizens and Syrian refugees.
Response: Done as suggested. Please note revised introduction section.
- Methods- can the authors better described how the cohort has been collected- was it pure random sample of individuals? What do they mean by snowball?
Response: Recruitment methodology was based on a convenient sampling strategy as stated in the original manuscript. For clarity, the snowball methodology is now explained in the revised manuscript.
- Results- Table 1- given the large numbers of births it’s a bit surprising that so few live with more than one family member.
Response: We apologize for lack of clarity. This is now revised. The statement in table 1 “live with more than one family” was replaced with “More than one family lives in the same house” to reflect the intended meaning.
- Discussion- obesity does slightly increase the risk for BC, as well as multi-parity, however the Syrian refugees who had more children performed less self-exam. Therefore, I would suggest to not link btw BC risk factors and adherence to screening. (Same applies for smoking at the end of the paragraph). This study does not have the power to link the 2.
Response: We agree with the reviewer, we have now removed this paragraph from the discussion section and have also eliminated corresponding correlation data form Table 3.
- Breast self-exam has been proved of not being an effective method of BC early detection, it can merely serve here as a surrogate for general awareness. This might be worth mentioning in the paper.
Response: The effectiveness of self-exam (BSE) on early diagnosis is debatable, several studies suggest the effectiveness of BSE on early diagnosis (e.g: PMID: 30547842, PMID: 3054784). However, we agree that it is not an indicator of BC awareness. We highlighted this issue in the revised manuscript.
- To the limitations of the study I suggest adding the fact that the average age was quite low, as women under 30, and especially under 25 are largely not the target population for BC screening, according to most European and American guidelines.
Response: This is an excellent suggestion. We have added this as a limitation.
- Unfortunately the full questionnaire was not part of the review. It’s interesting to see what were the barriers for screening questions.
Response: The questionnaires used in the present study were previously published and have been validated in Arabic language for Arabic speaking women, as indicated in the Methods section.